# A split fluorescent reporter with rapid and reversible complementation

Alison G. Tebo [1] & Arnaud Gautier [1]

Interactions between proteins play an essential role in metabolic and signaling pathways, cellular processes and organismal systems. We report the development of splitFAST, a fluorescence complementation system for the visualization of transient protein-protein interactions in living cells. Engineered from the fluorogenic reporter FAST (Fluorescence-Activating and absorption-Shifting Tag), which specifically and reversibly binds fluorogenic hydroxybenzylidene rhodanine (HBR) analogs, splitFAST displays rapid and reversible complementation, allowing the real-time visualization of both the formation and the dissociation of a protein assembly.

[1] PASTEUR, Department of Chemistry, École Normale Supérieure, PSL University, Sorbonne University, CNRS, 75005 Paris, France. Correspondence and requests for materials should be addressed to A.G. (email: arnaud.gautier@ens.fr)

Protein–protein interactions (PPIs) play a central role in most cellular processes. Our understanding of the molecular basis of biological functions depends on our ability to identify interacting partners and characterize how they interact in space and time. Because of their ability to provide spatial and temporal information, imaging tools based on Föster Resonance Energy Transfer (FRET)[1] or bimolecular fluorescence complementation (BiFC)[2] have become the tools of choice to study PPI.

BiFC assays are often preferred to FRET because they are easy to implement, straightforward to interpret, and less sensitive to the relative levels of the two interacting proteins. In BiFC assays, two proteins are fused to two complementary fragments of a fluorescent protein (FP), which assemble into a functional reporter if the two proteins interact together. As the two complementary fragments are not fluorescent when taken separately, high contrast is obtained no matter the relative proportion of the two interacting proteins. However, monitoring PPIs with BiFC has posed its own challenges. First, spontaneous self-assembly can generate unspecific fluorescence background. Furthermore, for BiFC based on proteins of the GFP family, complementation is followed by chromophore maturation, which results in irreversible complex formation[3], while for BiFC based on phytochrome-based infrared FPs, the attachment of the biliverdin chromophore is slow and often also results in irreversibility[4,5]. The slow formation of fluorescent complex prevents the monitoring of transient PPIs and the performance of dynamic studies involving active and inactive states, and may induce dominant-negative or -positive effects.

Here we present splitFAST, a reversible split fluorescent reporter that allows real-time monitoring of both formation and dissociation of a protein assembly (Fig. 1a). This split system was engineered from the fluorescence-activating and absorption shifting tag (FAST), a small protein of 14 kDa that specifically and reversibly binds hydroxybenzylidene rhodanine (HBR) analogs displaying various spectral properties[6–8]. HBR analogs are weakly fluorescent in solution, but strongly fluoresce when immobilized in the binding cavity of FAST. This fluorogenic behavior provides high contrast even in the presence of an excess of fluorogenic chromophore (the so-called fluorogen). In addition to providing many of the advantages of FPs for live cell imaging, FAST is characterized by very rapid fluorogen binding and thus fluorescence maturation, and a full reversibility of the FAST:fluorogen complex.

## Results

**splitFAST design**. With rapid and reversible fluorogen binding, FAST possessed all the qualities for the design of a split reporter displaying both rapid and reversible complementation. We split FAST in its last loop between Ser 114 and Gly 115 (Fig. 1a). The two fragments 1–114 (hereafter called as NFAST) and 115–125 (hereafter called as CFAST11) showed modest affinity in the presence of HMBR (4-hydroxy-3-methylbenzylidene rhodanine, which provides green-yellow fluorescence) or HBR-3,5DOM (4-hydroxy-3,5-dimethoxybenzylidene rhodanine, which provides orange-red fluorescence) (Supplementary Table 1 and Supplementary Fig. 1). The excitation and emission spectra of the complemented splitFAST:fluorogen assembly were identical to that of regular FAST:fluorogen complex (Supplementary Fig. 2). The apparent affinity of the two fragments could be further decreased by the successive removal of residues at the C terminus of CFAST11 (giving CFAST10-8) (Supplementary Table 1 and Supplementary Fig. 1).

**Observation of protein complex formation and dissociation**. To test the ability of splitFAST to detect PPIs in mammalian cells

(pretreated with HBR analogs), we fused NFAST and CFAST$n$ ($n = 10$ or 11) to the FK506-binding protein (FKBP) and the FKBP-rapamycin-binding domain of mammalian target of rapamycin (FRB) that interact together in the presence of rapamycin[9]. Rapamycin-induced FRB-FKBP dimerization led to a large fluorescence increase, in accordance with interaction-dependent complementation of splitFAST (Fig. 1b, d). With HMBR, the fluorescence fold increase upon association was in average 5-fold with CFAST11 and 12-fold with CFAST10 (Fig. 1b, d). The difference of dynamic range observed between CFAST11 and CFAST10 can be explained by the difference of affinity of CFAST11 and CFAST10 for NFAST: the lower affinity of CFAST10 for NFAST leads to less self-complementation, and thus to greater dynamic range. Self-complementation varied with protein expression level, but was accordingly in average 20% with CFAST11, and <10% with CFAST10. The use of HBR-3,5DOM gave similar results (Fig. 1c, d), demonstrating that the color of splitFAST could be tuned by changing the nature of the fluorogen added. In cells, splitFAST:fluorogen assemblies (with either CFAST11 or CFAST10) were furthermore shown to be as photostable as regular FAST:fluorogen (Supplementary Fig. 3). Time-lapse imaging after rapamycin addition showed fluorescence saturation within a few minutes, in agreement with the rapid formation of the FRB–FKBP–rapamycin complex (Fig. 1e and Supplementary Figs. 4a, 5a). The kinetics of assembly was independent of the CFAST$n$ or fluorogen used. Although we observed cell-to-cell variability in the time delay before seeing assembly likely due to cell-to-cell variability in the rate of diffusion of rapamycin across cell membrane, once association started, complementation was complete within few minutes. Overall, this set of experiments demonstrated that splitFAST can monitor protein complex formation in real time.

To test the reversibility of splitFAST, we used the ability of rapamycin to dissociate AP1510-induced FKBP homodimer[10]. Cells co-expressing FKBP-NFAST and FKBP-CFAST$n$ ($n = 10$ or 11) were incubated with AP1510 for 2 h to preform the FKBP homodimer and treated with HMBR. The addition of rapamycin led to a loss of splitFAST fluorescence of in average 80–90% in agreement with FKBP homodimer dissociation (Fig. 1f–h), demonstrating the reversibility of splitFAST assembly. Rapid loss of fluorescence within a few minutes was observed after rapamycin addition, demonstrating the rapid disassembly of splitFAST when two proteins dissociate (Fig. 1i and Supplementary Figs. 4b, 5b). As was observed for the association, we observed the same kinetics of disassembly no matter the CFAST$n$ or fluorogen used, although cell-to-cell variability in the time delay before dissociation starts was also observed, likely due to cell-to-cell variability in the rate of rapamycin diffusion.

The ability of splitFAST to image dynamic and reversible PPIs was further demonstrated by monitoring in a single experiment: first, the association of FKBP-NFAST and FKBP-CFAST$n$ ($n = 11$ or 10) upon addition of AP1510, and then, the dissociation of the FKBP-FKBP homodimer by removal of AP1510 and addition of rapamycin (Fig. 1j, k, Supplementary Fig. 6, Supplementary Movie 1, and Supplementary Movie 2).

**Detection of PPIs at the plasma membrane**. As many PPIs occur at the plasma membrane, we next asked if splitFAST could detect the interaction between a membrane protein and a cytosolic protein. We expressed FKBP-CFAST$n$ ($n = 11$ or 10) in the cytosol and FRB-NFAST at the plasma membrane using a Lyn11 membrane-anchoring sequence (Lyn11-FRB-NFAST). The addition of rapamycin led to the rapid formation of fluorescent splitFAST at the plasma membrane in HMBR-treated cells

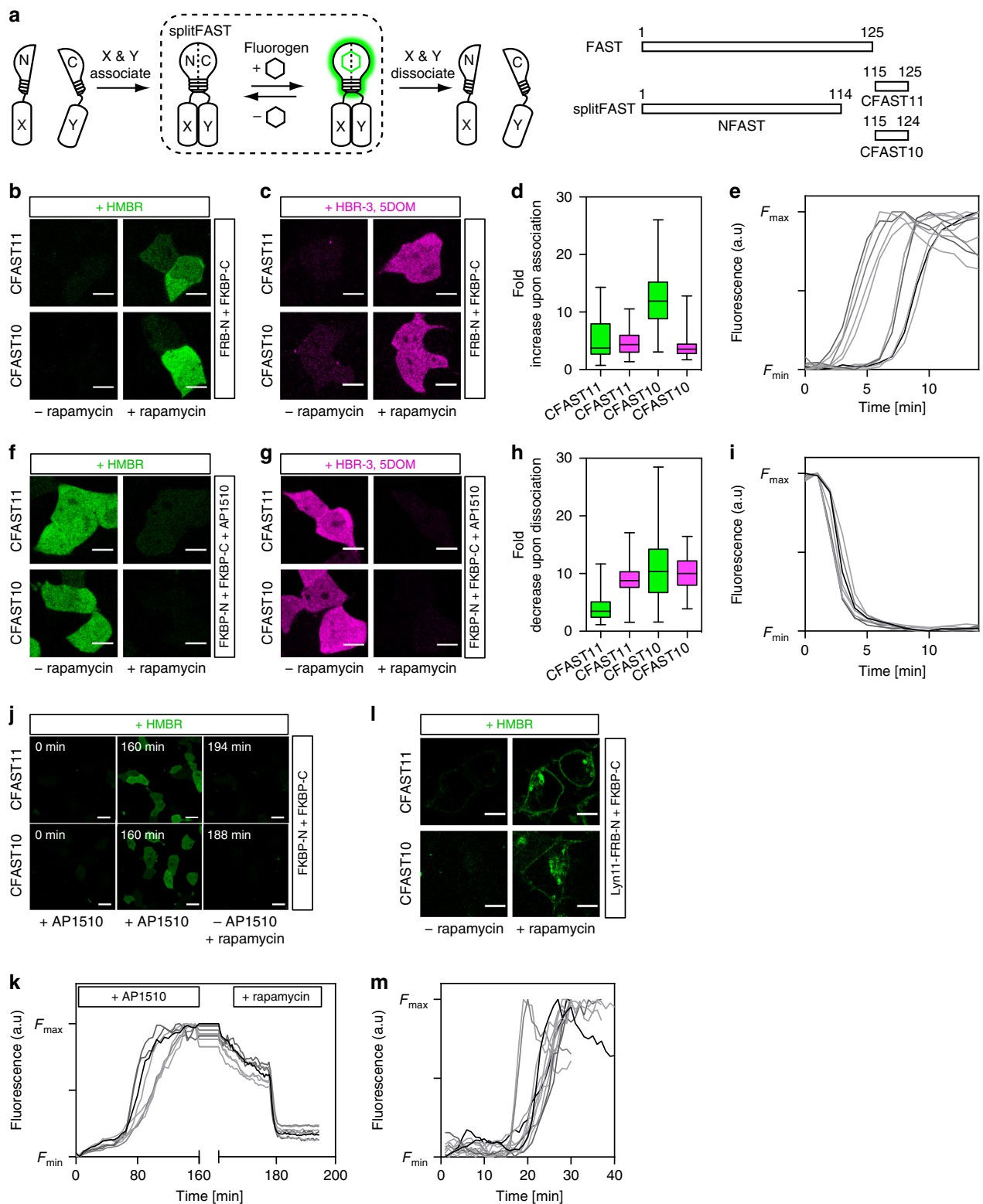

(Fig. 1l, m and Supplementary Fig. 7), demonstrating its ability to sense PPIs at the plasma membrane in real time.

**Detection of interactions from the MAPK signaling pathway.** We then benchmarked splitFAST with known, physiologically relevant PPIs from the mitogen-activated protein kinase (MAPK) signaling pathway. NFAST was fused to K-Ras, a small GTPase

downstream of growth factor receptors, and CFAST10 was fused to an mCherry fusion of Raf1, known to be recruited to the membrane by interaction with K-Ras[11,12]. The fluorescence of splitFAST was colocalized with that of mCherry and concentrated at the membrane, in agreement with a specific recruitment of Raf1 at the plasma membrane by K-Ras (Fig. 2a and Supplementary Fig. 8). Next, we looked at the interaction between the MAP kinase kinase MEK1 and the downstream extracellular

**Fig. 1** Rapid and reversible complementation allows the observation of transient protein–protein interactions. **a** SplitFAST principle and design. **b–e** HEK293T cells co-expressing FK506-binding protein (FKBP)-CFASTn ($n = 10$ or 11) and FKBP-rapamycin-binding domain of mammalian target of rapamycin (FRB)-NFAST were labeled with 5 μM HMBR (4-hydroxy-3-methylbenzylidene rhodanine) (**b**) or 10 μM HBR-3,5DOM (4-hydroxy-3,5-dimethoxybenzylidene rhodanine) (**c**) and imaged before and after the addition of 100 nM rapamycin. Scale bars 10 μm. **d** Fluorescence fold increase upon FKBP-FRB association: box plot with $n = 84$, 83, 107, and 112 cells, respectively, from three to four experiments (whiskers represent the highest and lowest values); green = HMBR; magenta = HBR-3,5DOM. **e** Temporal evolution of the fluorescence intensity after rapamycin addition in HMBR-treated cells co-expressing FRB-NFAST and FKBP-CFAST11 ($n = 11$ cells; see also Supplementary Figs. 4a, 5a). **f–i** AP1510-treated HEK293T cells co-expressing FKBP-NFAST and FKBP-CFASTn ($n = 10$ or 11) were labeled with 5 μM HMBR (**f**) or 10 μM HBR-3,5DOM (**g**) and imaged before and after the addition of 1 μM rapamycin. Scale bars 10 μm. **h** Fluorescence fold decrease upon FKBP-FKBP dissociation: box plot with $n = 142$, 175, 219, and 125 cells, respectively, from three to four experiments (whiskers represent the highest and lowest values); green = HMBR; magenta = HBR-3,5DOM. **i** Temporal evolution of the fluorescence intensity after rapamycin addition in AP1510-treated cells co-expressing FKBP-NFAST and FKBP-CFAST11 ($n = 8$ cells; see also Supplementary Figs. 4b, 5b). **j** HMBR-labeled cells co-expressing FKBP-NFAST and FKBP-CFASTn ($n = 11$ or 10) were firstly treated with 100 nM AP1510 for 160 min, then AP1510 was removed, and 1 μM rapamycin was added. Selected frames are shown (see also Supplementary Fig. 6, Supplementary Movie 1, and Supplementary Movie 2). Scale bars 30 μm. **k** Temporal evolution of the fluorescence intensity upon sequential treatment of HMBR-labeled cells co-expressing FKBP-NFAST and FKBP-CFAST11 with AP1510 and then rapamycin ($n = 8$ cells; see also Supplementary Fig. 6). **l** HEK293T cells co-expressing Lyn11-FRB-NFAST and FKBP-CFASTn ($n = 10$ or 11) were labeled with 5 μM HMBR and imaged before and after the addition of 100 nM rapamycin. Scale bars 10 μm. **m** Temporal evolution of the fluorescence intensity after rapamycin addition in HMBR-treated cells co-expressing Lyn11-FRB-NFAST and FKBP-CFAST11 ($n = 9$ cells; see also Supplementary Fig. 7)

signal-regulated protein kinase ERK2, one of the central interactions in the Raf/MEK/ERK signaling pathway. When MEK1 was fused to NFAST (MEK1-NFAST) and an mCherry fusion of ERK2 to CFAST10 (mCherry-ERK2-CFAST10), we observed specific, cytosolic splitFAST fluorescence in accordance with MEK1 anchoring ERK2 in the cytosol[13,14] (Fig. 2a and Supplementary Fig. 8). Finally, splitFAST allowed us to detect the nuclear interaction between ERK2 and MKP1 (DUSP1), which is a phosphatase localized in the nucleus responsible for deactivating ERK2 after its activation and subsequent translocation to the nucleus[15] (Fig. 2a and Supplementary Fig. 8).

**Real-time monitoring of the MEK1-ERK2 interaction**. To explore the applicability of splitFAST to study dynamic PPIs, we followed the interaction between MEK1 and ERK2 upon activation of the MAPK signaling pathway. Upon cell stimulation, MEK1 phosphorylates ERK2, which detaches from MEK1 and translocates to the nucleus[13,16], where it regulates the activity of transcription factors. Dephosphorylation by nuclear phosphatases deactivates ERK2, returning it to the cytoplasm[15,17,18]. In resting, HMBR-treated cells expressing MEK1-NFAST and mCherry-ERK2-CFAST10, and mCherry and splitFAST fluorescence were cytoplasmic, in agreement with MEK1 anchoring ERK2 in the cytoplasm (Fig. 2b, d and Supplementary Movie 3). Upon cell stimulation with epidermal growth factor (EGF), mCherry-ERK2-CFAST10 dissociated from MEK1-NFAST and translocated to the nucleus, as shown by the simultaneous loss of splitFAST fluorescence and the nuclear accumulation of mCherry fluorescence (Fig. 2b, d and Supplementary Movie 3). The nuclear accumulation of mCherry-ERK2-CFAST10 was transitory: desensitized mCherry-ERK2-CFAST10 returned to the cytoplasm and re-assembled with MEK1-NFAST, as revealed by the simultaneous increase of splitFAST fluorescence and cytosolic mCherry fluorescence (Fig. 2b, d and Supplementary Movie 3). This experiment illustrated how splitFAST enabled us to observe dynamic PPIs in signaling pathways in real time.

**Real-time monitoring of transient Ca²⁺-dependent interactions**. To further demonstrate the use of splitFAST for the detection of rapid and transient interactions, we next monitored the Ca²⁺-dependent interaction between calmodulin (CaM) and the Ca²⁺-CaM-interacting peptide M13. In HMBR-treated HeLa cells expressing CFAST10-CaM and M13-NFAST, the addition of histamine led to a large increase of fluorescence, followed by rapid oscillations of the fluorescence signals and eventually

desensitization (Fig. 2e–g and Supplementary Movie 4). We observed oscillations as short as 30–60 s, in agreement with the known change in Ca²⁺ concentration in mammalian cells upon histamine stimulation observed with gold standard Ca²⁺ indicators such as Fura-2[19]. This experiment showed that split-FAST could serve as reporting module for the design of complementation-based biosensors able to monitor rapid transient events.

**splitFAST-based apoptosis biosensor**. To examine the utility of splitFAST for imaging other signaling processes, we created a caspase biosensor. We fused the transcriptional regulator bFos to CFAST (bFos-CFAST11), and constructed a gene encoding bJun-NFAST-NLS₃-DEVDG-mCherry-NES, where NES is a genetically fused nuclear export signal, DEVD is the caspase-3 substrate sequence Asp-Glu-Val-Asp, NLS is a nuclear localization signal, and bJun is a peptide known to form a heterodimer with bFos[20,21] (Fig. 2h). Induction of apoptosis (and thus caspase-3 activity) by treatment with staurosporine released bJun-NFAST from mCherry-NES, resulting in a translocation of bJun-NFAST to the nucleus, and the subsequent complementation of splitFAST by interaction of bJun and bFos. Approximately 1–2 h after the induction of apoptosis, we observed the segregation of red fluorescence in the cytoplasm and the appearance of the bright green fluorescence of complemented splitFAST in the nucleus (Fig. 2h–j and Supplementary Movie 5). Beyond further demonstrating the potential of splitFAST to monitor PPI formation in real time, this experiment showed the great potential of splitFAST for the design of cellular biosensors.

**Discussion**

The presented results demonstrate that splitFAST can be a versatile tool for studying PPIs and designing new biosensors. SplitFAST is adaptable to multi-color imaging, as it can fluoresce green-yellow or orange-red light depending on the fluorogen used. The dynamic properties of splitFAST can be further tuned to the experimental context using either CFAST11 or CFAST10. We recommend to systematically test both versions as complementation efficiency and dynamic range can vary from interaction to interaction and can depend on various parameters such as the expression level of the two proteins, their cellular localizations, the relative positioning, and orientation of the two split fragments within the protein assembly or the linkers used. Moreover, as for all tag-based methods, systematic verification of the function of the fusions is recommended.

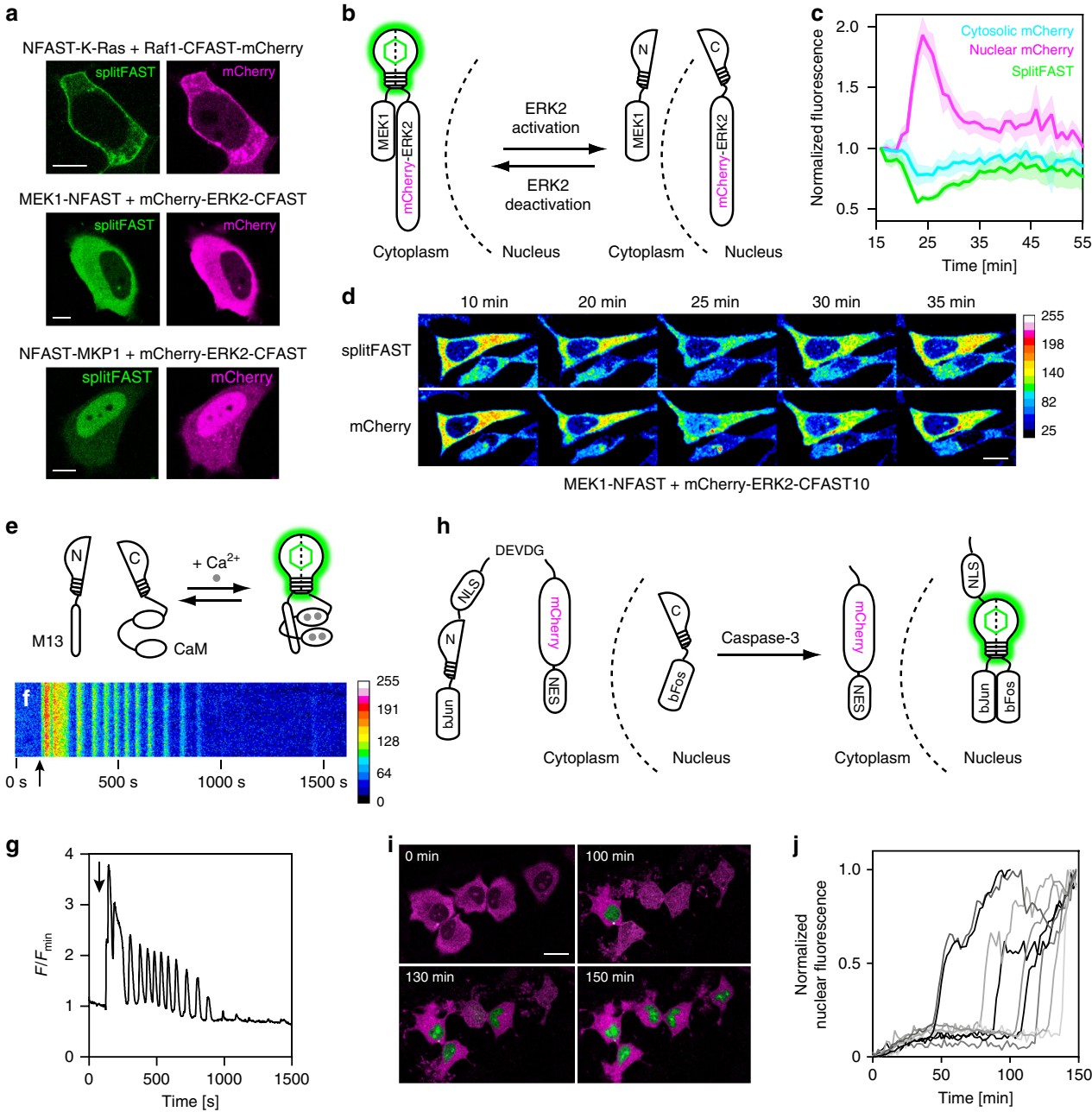

**Fig. 2** Use of splitFAST for imaging protein–protein interactions and intracellular signaling. **a** Use of splitFAST for imaging K-Ras/Raf1, MEK1/ERK2, and ERK2/MKP1 interactions. Representative images of cells co-expressing the indicated constructs were imaged in the presence of 10 μM HMBR (4-hydroxy-3-methylbenzylidene rhodanine) (Supplementary Fig. 8 shows control experiments). **b** Use of splitFAST for imaging the evolution of MEK1/ERK2 interaction upon epidermal growth factor (EGF) stimulation. HMBR-labeled HeLa cells co-expressing MEK1-NFAST and mCherry-ERK2-CFAST10 were imaged after stimulation with EGF. **c** Temporal evolution of splitFAST fluorescence (green), cytoplasmic mCherry fluorescence (cyan), and nuclear mCherry fluorescence (magenta) intensities (mean ± s.e.m., $n = 6$ cells, five experiments). Data were synchronized using the beginning of the nuclear import of mCherry-ERK2-CFAST10 as reference. **d** Selected frames of a representative cell (see also Supplementary Movie 3). Scale bar 10 μm. **e–g** Use of splitFAST for imaging of the $Ca^{2+}$-dependent interaction of calmodulin (CaM) and the $Ca^{2+}$-CaM interacting peptide M13. **e** The sensor is composed of M13-NFAST and CFAST10-CaM. **f**, **g** Temporal evolution of the intracellular fluorescence intensity for a representative HeLa cell ($n = 14$ cells from two experiments) treated with histamine (histamine addition is shown by the arrow) (see also Supplementary Movie 4). **h–j** Use of splitFAST for detecting caspase-3 activity. **h** Sensor design. **i** Selected merged frames from two-color imaging of representative HMBR-labeled cells ($n = 30$ cells from three experiments) expressing bFos-CFAST11 and bJun-NFAST-NLS3-DEVDG-mCherry-NES after treatment with staurosporine (see also Supplementary Movie 5). The mCherry signal is in magenta, while the splitFAST signal is in green. Scale bar 20 μm. **j** Temporal evolution of the nuclear fluorescence intensity ($n = 9$ cells)

To conclude, we develop splitFAST, a split reporter displaying rapid and reversible complementation, allowing one to observe transient PPIs in real time. SplitFAST allows the observation of PPIs in various cellular compartments (cytosol, nucleus, plasma membrane). In contrast to traditional BiFC systems, splitFAST complementation is fully reversible and disassembly is rapid, which allows not only the real-time monitoring of protein complex assembly but also the real-time monitoring of protein complex disassembly. This unprecedented behavior opens exciting prospects to study the role and function of PPIs in various cellular processes and dissect complex interaction networks.

## Methods

**General**. Synthetic oligonucleotides used for cloning were purchased from Sigma-Aldrich or Integrated DNA Technology. The sequences of oligonucleotides used in this study are provided in Supplementary Table 2. PCR reactions were performed with Q5 polymerase (New England Biolabs) in the buffer provided. PCR products were purified using QIAquick PCR purification kit (Qiagen). The products of restriction enzyme digests were purified by preparative gel electrophoresis followed by QIAquick gel extraction kit (Qiagen). Restriction endonucleases, T4 ligase, Phusion polymerase, Taq ligase, and Taq exonuclease were purchased from New England Biolabs and used with accompanying buffers and according to the manufacturer's protocols. Isothermal assemblies (Gibson assembly) were performed using homemade mix prepared according to previously described protocols[22]. Small-scale isolation of plasmid DNA was done using QIAprep miniprep kit (Qiagen) from 2 mL of overnight culture. Large-scale isolation of plasmid DNA was done using the QIAprep maxiprep kit (Qiagen) from 150 mL of overnight culture. All plasmid sequences were confirmed by Sanger sequencing with appropriate sequencing primers (GATC-Biotech). Please see Supplementary Table 3 for a list of all plasmids used in this study. Peptides corresponding to CFAST11-8 were purchased from Cliniciences at 98% purity and are acetylated and amidated at the N and C termini. Rapamycin was purchased from Sigma-Aldrich and dissolved in dimethyl sulfoxide to a concentration of 3 mM. AP1510 was purchased from Clontech and dissolved in ethanol to a concentration of 0.5 mM. Human recombinant EGF was purchased from Sigma-Aldrich and dissolved in 0.1% bovine serum albumin to a concentration of 100 µg/ml. Staurosporine was purchased from Cell Signaling Technologies and dissolved in ethanol to a concentration of 1 mM. The preparation of HMBR and HBR-3,5DOM was previously described[6,7]. HMBR and HBR-3,5DOM are available from The Twinkle Factory under the name $^{TF}$Lime and $^{TF}$Coral (thetwinklefactory.com).

**Sequences**. Protein sequence of NFAST: MEHVAFGSEDIENTLAKMDDGQLD GLAFGAIQLDGDGNILQYNAAEGDITGRDPKQVIGKNFFKDVAPGTDSPEFY GKFKEGVASGNLNTMFEWMIPTSRGPTKVKVHMKKALS

Protein sequence of CFAST11: GDSYWVFVKRV

Protein sequence of CFAST10: GDSYWVFVKR

DNA sequence of NFAST: atggagcatgttgcctttggcagtgaggacatcgagaacactctggcca aaatggacgacggacaactggatgggttggcctttggcgcaattcagctcgatggtgacgggaatatcctgcagtacaat gctgctgaaggagacatccaggcagagatcccaaacaggtgattgggaagaacttcttcaaggatgttgcacctggaac ggattctcccgagttttacggcaaattcaaggaaggcgtagccgtcagggaatctgaacaccatgttgaatggatgatacc gacaagcaggggaccaaccaaggtcaaggtgcacatgaagaaagccctttcc

DNA sequence of CFAST11: ggtgacagctattgggtctttgtgaaacgggtg

DNA sequence of CFAST10: ggtgacagctattgggtctttgtgaaacgg

**Molecular cloning**. *Bacterial expression plasmids*: The plasmid pAG209 was obtained by inserting the gene encoding for NFAST (amplified using primers ag126 and ag216) into plasmid pET28a using restriction enzymes *NheI* and *XhoI*.

*FRB-FKBP fusion plasmids for mammalian expression*: In general, fusions were constructed by PCR assembly and contain an 11 amino acid linker, SGGGGSGGGGS, between the two proteins. The plasmid pAG148 was obtained by inserting the gene encoding FRB-NFAST (the sequence coding for FRB-NFAST was assembled by PCR from the sequences coding for FRB and NFAST amplified with the primers ag181/ag182 and ag175/ag176) into plasmid pAG104[6] using the restriction enzymes, *BglII* and *NotI*. The plasmid pAG149 was generated by inserting the gene encoding FKBP-NFAST (the sequence coding for FKBP-NFAST was assembled by PCR from the sequences coding for FKBP and NFAST amplified with the primers ag183/ag184 and ag175/ag176) into plasmid pAG104 using the restriction enzymes *BglII* and *NotI*. The plasmid pAG152 was obtained by inserting the gene FRB-CFAST11 (synthesized by Eurofins Genomics) into pAG104 using restriction enzymes *BglII* and *NotI*. The plasmid pAG153 was generated by inserting the gene for FKBP (amplified using primers ag183 and ag184) into pAG152 using restriction enzymes *BglII* and *BspEI*.

The plasmid pAG241 was constructed by Gibson assembly of two fragments obtained by amplification of the plasmid pAG153 with the primers ag345/ag313 and ag347/ag314. The plasmid pAG336 was cloned by Gibson assembly of two fragments obtained by amplification of the plasmid pAG148 with the primers

ag468/313 and 469/314. To determine the photostability of splitFAST in cell, the plasmid pAG439 encoding FRB-NFAST-P2A-mCherry was generated by Gibson assembly of the sequences of FRB-NFAST (amplified from pAG148 with primers ag308 and ag313) and mCherry (amplified from pAG96[6] with ag412 and ag550) assembled with the plasmid backbone of pAG104 (amplified using primers ag347 and ag314). The plasmid pAG496 encoding FKBP-CFAST10-IRES-mCherry was cloned from the plasmid pAG241 (amplified using ag700 and ag313), mCherry (amplified using ag701 and ag314), and the IRES sequence amplified from the plasmid pIRES (using primers ag694 and ag695) via Gibson assembly. The plasmid pAG490 encoding FRB-NFAST-IRES-mTurquoise2 was cloned from the plasmid pAG148 (amplified using ag697 and ag313), the IRES sequence amplified from the plasmid pIRES (using ag694 and ag695), and a g-block encoding mTurquoise2 (IDT). The plasmid pAG491 encoding lyn11-FRB-NFAST-IRES-mTurquoise2 was cloned from the plasmid pAG336 (amplified using ag697 and ag313), the IRES sequence amplified from the plasmid pIRES (using ag694 and ag695), and a g-block encoding mTurquoise2 (IDT).

*Signaling pathway plasmids*: The genes for NFAST-MKP1, NFAST-KRas, and CFAST10-Raf1-mCherry were synthesized by Eurofins genomics. The plasmids pAG301, pAG341, and pAG340 were generated via Gibson assembly of the sequences of NFAST-MKP1 (amplified using ag357/ag412), NFAST-KRas (amplified using ag357/ag475), and CFAST10-Raf1-mCherry (amplified using ag474/ag412), and the backbone of pAG104 amplified in two fragments using primers ag311/ag314 and ag358/ag313. The genes of MEK1, ERK2, and mCherry were amplified using primers ag418/419, ag416/417, and ag414/415, respectively, and assembled via Gibson assembly with the corresponding fragments of NFAST (amplified with primers 374/313) and CFAST10 (amplified with primers 413/314) to generate the plasmids pAG298 and pAG296 encoding NFAST-MEK1 and mCherry-ERK2-CFAST10, respectively. The plasmids pAG492, pAG493, and pAG494 were constructed by Gibson assembly from the initial plasmid encoding the signaling pathway partner pAG298, pAG301, pAG341 (amplified with primers 697/313, 696/314, 699/313, 698/313, the IRES sequence (amplified from the plasmid pIRES using ag694 and ag695)), and a g-block encoding mTurquoise2 (IDT).

*Sensor construction*: The plasmid pAG334 was generated from the plasmid pAG148 by Gibson assembly by amplifying NFAST using primers ag455/ag456 with M13 encoded, and the backbone of pAG148 with primers ag313/ag314. The plasmid pAG335 was generated by Gibson assembly by amplification of CaM from a previously published plasmid[23] using primers ag465 and ag466, and inserted into the plasmid pAG144 by amplifying CFAST10 with primers ag467 and ag313 and assembling with plasmid amplified with ag311 and ag314. The genes encoding for bJun-NFAST and bFos-CFAST11 were synthesized by Eurofins genomics. The plasmid pAG384 was cloned by Gibson assembly by amplification of the gene encoding for bFos-CFAST11 using primers ag541/542 and the backbone of pAG104 using primers ag358/ag313 and ag347/ag314. The plasmid pAG385 was generated by amplifying the gene for bJun-NFAST using primers ag543/544, followed by a Gibson assembly with the sequence of mCherry amplified with ag545/546, the primer ag535 encoding for NLSx3 and the backbone of pAG104 amplified using the primers ag358/313 and ag539/ag314.

**Protein expression and purification**. Expression vectors were transformed in Rosetta (DE3) pLysS *Escherichia coli* (New England Biolabs). Cells were grown at 37 °C in LB medium complemented with 50 µg/ml kanamycin and 34 µg/ml chloramphenicol to OD$_{600nm}$ 0.6. Expression was induced for 4 h by adding isopropyl β-D-1-thiogalactopyranoside to a final concentration of 1 mM. Cells were harvested by centrifugation (4000 × *g* for 20 min at 4 °C) and frozen. The cell pellet was resuspended in lysis buffer (phosphate buffer 50 mM, NaCl 150 mM, MgCl$_2$ 2.5 mM, protease inhibitor, DNase, pH 7.4) and sonicated (5 min at 20 % of amplitude, 3 s on, 1 s off). The lysate was incubated for 2 h at 4 °C to allow DNA digestion by DNase. Cellular fragments were removed by centrifugation (9200 × *g* for 1 h at 4 °C). The supernatant was incubated overnight at 4 °C under gentle agitation with Ni-NTA agarose beads in phosphate-buffered saline (PBS) (sodium phosphate 50 mM, NaCl 150 mM, pH 7.4) complemented with 10 mM imidazole. Beads were washed with 20 volumes of PBS containing 20 mM imidazole, and with 5 volumes of PBS complemented with 40 mM imidazole. His-tagged proteins were eluted with 5 volumes of PBS complemented with 0.5 M imidazole. The buffer was exchanged to PBS (50 mM phosphate, 150 mM NaCl, pH 7.4) using PD-10 desalting columns.

**Physico-chemical measurements**. Steady-state ultraviolet–visible (UV–Vis) absorption spectra were recorded using a Cary 300 UV–Vis spectrometer (Agilent Technologies), equipped with a Versa20 Peltier-based temperature-controlled cuvette chamber (Quantum Northwest), and fluorescence data were recorded using a LPS 220 spectrofluorometer (PTI, Monmouth Junction, NJ, USA), equipped with a TLC50TM Legacy/PTI Peltier-based temperature-controlled cuvette chamber (Quantum Northwest).

Thermodynamic dissociation constants for NFAST:CFAST$n$ ($n = 8–11$) couples were determined using peptides synthesized for CFAST$n$ ($n = 8–11$) and recombinantly purified NFAST. The affinity for NFAST:CFAST11 in the

presence of 10 μM HMBR was determined independently from a minimum of three different purifications of NFAST. NFAST:CFAST11 was then run in parallel as an internal control for the determination of the other NFAST-CFAST combinations, which were all performed on the same day with the same preparation of NFAST. Thermodynamic dissociation constants were determined with a Spark 10M plate reader (Tecan) and fit in Prism 6 to a one-site-specific binding model.

**Mammalian cell culture.** HEK 293T cells (ATCC CRL-3216) were cultured in Dulbecco's modified Eagle's medium (DMEM) supplemented with phenol red, Glutamax I, and 10% (vol/vol) fetal calf serum, at 37 °C in a 5% $CO_2$ atmosphere. HeLa cells (ATCC CCL-2) were cultured in modified Eagle's medium (MEM) supplemented with phenol red, 1× non-essential amino acids, 1× sodium pyruvate, and 10% (vol/vol) fetal calf serum at 37 °C in a 5% $CO_2$ atmosphere. For imaging, cells were seeded in μDish IBIDI (Biovalley) coated with poly-L-lysine. Cells were transiently transfected using Genejuice (Merck) or Lipofectamine 2000 (Invitrogen) according to the manufacturer's protocol for 24 h prior to imaging.

**Fluorescence microscopy.** Confocal micrographs were acquired on a Zeiss LSM 710 Laser Scanning Microscope equipped with a Plan Apochromat ×63/1.4 NA oil DIC M27 immersion objective, heated stage, and XL-LSM 710 S1 incubation chamber for temperature and $CO_2$ control. Images were acquired using ZEN software and processed in Fiji (ImageJ). The same absolute color scaling was used when side-by-side comparisons and time lapses are shown.

Photobleaching measurements for HMBR were carried out at 10 μM fluorogen at 488 nm excitation (4.6 kW/cm$^2$, 1.27 μs pixel dwell); FAST was used as a control. Samples were imaged continuously for 1000 images at a 1 frame/s frequency.

To image the rapamycin-mediated interaction between FRB and FKBP, the cells were imaged in DMEM without phenol red supplemented with 5 μM HMBR or 10 μM HBR-3,5DOM. Tile images were taken before rapamycin addition. A solution of rapamycin, prepared in fluorogen-containing DMEM in order to maintain fluorogen concentration constant, was added to obtain a final rapamycin concentration of 100 nM and images were taken every minute. A final tile image was taken after fluorescence saturation. To image the AP1510-mediated interaction, AP1510 was added to a final concentration of 100 nM to the cells for 2 h before imaging. The cells were rinsed and the media were replaced with DMEM without phenol red, supplemented with 5 μM HMBR or 10 μM HBR-3,5DOM. Tile images were taken before rapamycin addition. A solution of rapamycin (prepared in DMEM supplemented with fluorogen in order to maintain fluorogen concentration constant) was added to obtain a final rapamycin concentration of 1 μM and images were taken every 30 s. A final tile image was taken after fluorescence ceased changing. To image the association and dissociation of FKBP-FKBP in the same sample, optiMEM was supplemented with 5 μM HMBR and AP1510 was added to a final concentration of 100 nM just before the beginning of imaging. The cells were maintained at 37 °C and 5% $CO_2$ over the duration of the experiment. Images were taken every 5 min until the fluorescence signal saturated. The acquisition was then paused and the sample was washed with 1× Dulbecco's PBS (supplemented with HMBR in order to maintain fluorogen concentration constant) and the imaging solution was replaced. The acquisition frequency was reduced to 30 s per image and 7–10 images were acquired before a solution of rapamycin, prepared in optiMEM (supplemented with HMBR in order to maintain fluorogen concentration constant), was added to obtain a final ramapycin concentration of 1 μM.

To image interactions in the Raf-MEK-ERK pathway, the cells were serum starved for 24 h before imaging after transfection. The cells were imaged in DMEM without phenol red supplemented with 10 μM HMBR. For time-course experiments, the pathway was activated using purified EGF added to a final concentration of 200 ng/ml.

Calcium imaging was performed in HHBSS supplemented with 5 μM HMBR. Calcium oscillations were triggered using 50 μM histamine in HHBSS (supplemented with HMBR in order to maintain fluorogen concentration constant) and images were acquired every 500 ms.

To image caspase activity, the cells were imaged at 37 °C in 5% $CO_2$ in optiMEM supplemented with 5 μM HMBR. Just before the start of acquisition, staurosporine was added to a final concentration of 2 μM. Images were acquired every 5 min over 3 h.

**Reporting summary.** Further information on research design is available in the Nature Research Reporting Summary linked to this article.

## Data availability

All relevant data are available from the authors.

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

## Acknowledgements

This work has been supported by the European Research Council (ERC-2016-CoG-724705 FLUOSWITCH), France BioImaging (ANR-10-INBS-04), and the Equipex Morphoscope 2 (ANR-11-EQPX-0029).

## Author contributions

A.G and A.G.T. designed and analyzed the experiments. A.G.T. performed the experiments. A.G. and A.G.T. wrote the paper.

## Additional information

**Competing interests:** A.G. is co-founder and holds equity in Twinkle Bioscience/The Twinkle Factory, a company commercializing the FAST technology. The other author declares no competing interests.

