## [Peer Review File · Nature Communications]

Reviewers' comments:

Reviewer #1 (Remarks to the Author):

The manuscript by Tebo et al describes the construction and application of splitFAST, a system for the monitoring of reversible interactions based on complementation and binding of an exogenous chromophore. In principle, such a system shares the advantages of the well-established BiFC methodology, but also addresses its most significant flaws: the slow rate of maturation, and the irreversibility of the association. The authors also demonstrate multiple applications, including the use of the system for the construction of biosensors.

Overall this contribution is of very high relevance to the large community interested in the monitoring of protein-protein interactions. The authors appear to have performed their experiments with care, and their conclusions appear to be justified by the data. As such publication in a high-impact journal such as Nature Communications is entirely appropriate.

I did identify some issues that the authors should address before publication:

- my biggest issue is that the graphs showing fluorescence time courses (e.g. Fig 1e) are all scaled between "Fmax" and "Fmin". This does not permit an assessment of the level of the background fluorescence. For example, suppose that the signal changes from 1 to 100 arbitrary units (low background), or from 90 to 100 a.u. (high background). In the latter case, there is a strong background or baseline fluorescence emission in the sample, and clearly this is much less desirable. However, the author's choice of formatting makes it difficult to distinguish between both scenarios. The authors should convincingly clarify this, and explicitly discuss the baseline/background emission and its origins (spontaneous complementation, emission from the unbound chromophore, background from other compounds in the cell). This can all be verified using appropriate control experiments.

- Can the authors shed more light on the kinetics of the association/dissociation process from their data? For example, Figure S4 suggests that CFAST-11 has a faster response to both dissociation and association events, though I found little discussion of this aspect in the manuscript. The calcium data, for instance, suggests very good temporal performance for the split-YFAST system, and I feel that the authors could highlight this more in the manuscript.

- Which construct (CFAST-11, CFAST-10, etc.) do the authors recommend for which experiments?

- Figure S4a: what are the bright spots in the lower left panels? (e.g. in CFAST-10 0 min). Why does the brightness of these spots change in time? The same absolute color scaling (LUT settings) must be used in all panels, so I just want to verify that this is indeed the case.

Reviewer #2 (Remarks to the Author):

Researchers who study dynamic protein-protein interactions would certainly benefit from an imaging tool that would allow measurement of trafficking and assembly/disassembly in real time. The authors have constructed such a tool, characterizing it with several attempts at optimization and with several known protein-protein interactions. The splitFAST tool is impressive and the paper was generally well written. The data are solid but I have a few concerns that could be addressed.

1) Using several examples, they demonstrate the system is in principle able to dynamically measure cycles of assembly and disassembly. It would have been helpful to demonstrate that the membrane-localized version was reversible in Figure 11,m.

2) They use several cytosolic/nuclear interactors to demonstrate the utility of splitFAST. They show that some of the interactions such as MEK1/ERK2 are reversible providing additional proof of concept that the system is robust. Can the authors compare calcium dynamics to the splitFAST assay directly in Figure 2e,f,g? Again, perhaps a membrane-localized interaction such as GPCR-

mediated recruitment of beta-arrestin could be added to bolster this aspect?

3) A general concern of all such tagged proteins is do they retain function when engineered with tags such as splitFAST. Can the authors comment on this?

Responses to reviewers

Reviewer #1 (Remarks to the Author):

The manuscript by Tebo et al describes the construction and application of splitFAST, a system for the monitoring of reversible interactions based on complementation and binding of an exogenous chromophore. In principle, such a system shares the advantages of the well-established BiFC methodology, but also addresses its most significant flaws: the slow rate of maturation, and the irreversibility of the association. The authors also demonstrate multiple applications, including the use of the system for the construction of biosensors.

Overall this contribution is of very high relevance to the large community interested in the monitoring of protein-protein interactions. The authors appear to have performed their experiments with care, and their conclusions appear to be justified by the data. As such publication in a high-impact journal such as Nature Communications is entirely appropriate.

#1 We thank reviewer 1 for his/her review and for acknowledging the significance of this work.

I did identify some issues that the authors should address before publication:

- my biggest issue is that the graphs showing fluorescence time courses (e.g. Fig 1e) are all scaled between "Fmax" and "Fmin". This does not permit an assessment of the level of the background fluorescence. For example, suppose that the signal changes from 1 to 100 arbitrary units (low background), or from 90 to 100 a.u. (high background). In the latter case, there is a strong background or baseline fluorescence emission in the sample, and clearly this is much less desirable. However, the author's choice of formatting makes it difficult to distinguish between both scenarios. The authors should convincingly clarify this, and explicitly discuss the baseline/background emission and its origins (spontaneous complementation, emission from the unbound chromophore, background from other compounds in the cell). This can all be verified using appropriate control experiments.

#2 The reason of displaying the graphs representing single-cell kinetic experiments scaled between Fmax and Fmin is to facilitate the comparison of the different experiments. Microscope images on Fig 1b,c,f,g,j,l, S4, S5, S6, S7 show representative change of fluorescence intensity upon association and dissociation, and allows, according to us, the readers to evaluate the dynamic range of the fluorescence change. Moreover, we present on fig 1d and 1h quantification of the fluorescence change upon association or dissociation. The change in fluorescence intensity is typically between 5-fold and 10-fold, which allows to assess the level of background, respectively 20% and 10%. The background originates from spontaneous self-complementation as the fluorogen itself displays little to no background. We have added a sentence in the main text page 5, and change one sentence on page 6, to clarify this point.

Page 5: “With HMBR, the fluorescence fold increase upon association was around 5-fold with CFAST11 and 12-fold with CFAST10 (**Figure 1b,d**). The difference of dynamic range observed between CFAST11 and CFAST10 can be explained by the difference of affinity of CFAST11 and CFAST10 for NFAST: the lower affinity of CFAST10 for NFAST leads to less self-complementation, and thus to greater dynamic range. Self-complementation varied with protein expression level, but was accordingly typically around 20% with CFAST11, and less than 10% with CFAST10.”

Page 6: “Addition of rapamycin led to a loss of splitFAST fluorescence of about 80-90% in agreement with FKBP homodimer dissociation (Figure 1f-h), demonstrating the reversibility of splitFAST assembly”.

- Can the authors shed more light on the kinetics of the association/dissociation process from their data? For example, Figure S4 suggests that CFAST-11 has a faster response to both dissociation and association events, though I found little discussion of this aspect in the manuscript.

#3 The kinetics of association and dissociation are roughly the same for CFAST11 and CFAST10. The observed differences between experiments are more due to cell-to-cell variability in the rate of diffusion of rapamycin across cell membrane, which then changes the time delay before seeing association or dissociation, rather than a difference in kinetics of association or dissociation. Furthermore, we have observed that the kinetics of the response are affected by the concentration of rapamycin, thus lending credence to the assertion that the rate of diffusion across the membrane is the rate-limiting factor in splitFAST response. The careful reader will note that one can follow the diffusion of rapamycin intracellularly, as the response is first seen in the cytosol before the nucleus. In the case of FRB:FKBP association, once the association starts, it is over within a few minutes, no matter the CFASTn or fluorogen used. Similar kinetics are observed for the dissociation of FKBP homodimer. We added sentences on pages 5 and 6 to specify this point.

Page 5: “The kinetics of assembly was independent of the CFASTn or fluorogen used. Although, we observed cell-to-cell variability in the time delay before seeing assembly likely due to cell-to-cell variability in the rate of diffusion of rapamycin across cell membrane, once association started, complementation was complete within a few minutes.”

Page 6: “As for association, we observed the same kinetics of disassembly no matter the CFASTn or fluorogen used, although cell-to-cell variability in the time delay before dissociation starts was also observed due likely to cell-to-cell variability in the rate of rapamycin diffusion.”

The calcium data, for instance, suggests very good temporal performance for the split-YFAST system, and I feel that the authors could highlight this more in the manuscript.

#4 We thank the reviewer 1 for the suggestion. We added a sentence on page 9 and change a sentence in the conclusion to better highlight the good temporal performance of splitFAST.

Page 9: “We observed oscillations as short as 30-60 seconds, in agreement with the known change in Ca^{2+} concentration in mammalian cells upon histamine stimulation observed with gold standard Ca^{2+} indicators such as Fura-2 (ref. 19). This experiment showed that splitFAST could serve as reporting module for the design of complementation-based biosensors able to monitor rapid transient events.”

Conclusion sentence change: “In contrast to traditional BiFC systems, splitFAST complementation is fully reversible and disassembly is rapid, which allows not only the real-time monitoring of protein complex assembly but also the real-time monitoring of protein complex disassembly.”

- Which construct (CFAST-11, CFAST-10, etc.) do the authors recommend for which experiments?

#5 We recommend to test systematically both as complementation efficiency and dynamic range can vary from interaction to interaction and can depend on various parameters including the expression level of the two proteins, the cellular localizations of the two proteins, the relative positioning/orientation of the two split fragments, the linkers used etc. We reshaped the conclusion to specify this point.

- Figure S4a: what are the bright spots in the lower left panels? (e.g. in CFAST-10 0 min). Why does the brightness of these spots change in time? The same absolute color scaling (LUT settings) must be used in all panels, so I just want to verify that this is indeed the case.

#6 The same absolute color scaling was used when time-lapses and side-by-side comparisons are shown. A sentence was added in the Materials and Methods part to specify this. The bright spots on FigS4a are simply due to cell debris (that tend to accumulate fluorogens) moving in and out the confocal plane. Note that although some debris are moving, some bright spots are also immobile and present in all the images.

Reviewer #2 (Remarks to the Author):

Researchers who study dynamic protein-protein interactions would certainly benefit from an imaging tool that would allow measurement of trafficking and assembly/disassembly in real time. The authors have constructed such a tool, characterizing it with several attempts at optimization and with several known protein-protein interactions. The splitFAST tool is impressive and the paper was generally well written. The data are solid but I have a few concerns that could be addressed.

#7 We thank reviewer 2 for his/her review and for acknowledging the significance of this work.

1) Using several examples, they demonstrate the system is in principle able to dynamically measure cycles of assembly and disassembly. It would have been

helpful to demonstrate that the membrane-localized version was reversible in Figure 1l,m.

#8 We thank reviewer 2 for this suggestion. As he/she acknowledged, we demonstrate on several systems that complementation is reversible. Given that membrane-associated FRB:FKBP dimer formation proceeds in the same fashion as cytosolic FRB:FKBP dimer formation, we do not expect the dissociation of an FKBP:FKBP dimer localized at the membrane to behave differently from what we observed in the cytosol. We thus do not believe that adding an experiment showing the dissociation of membrane-anchored FKBP:FKBP homodimer would add meaningful information as to the performance of the system. According to what is known on AP1510-induced FKBP-FKBP homodimer, nothing suggests that FKBP:FKBP homodimer disassembly would be affected if one monomer is anchored at the membrane.

2) They use several cytosolic/nuclear interactors to demonstrate the utility of splitFAST. They show that some of the interactions such as MEK1/ERK2 are reversible providing additional proof of concept that the system is robust. Can the authors compare calcium dynamics to the splitFAST assay directly in Figure 2e,f,g?

#9 As mentioned in our response **#4** to reviewer 1's comment, we have added on page 9 the following sentence to compare the response obtained with splitFAST and what was previously reported with gold standard calcium indicators.

“We observed oscillations as short as 30-60 seconds, in agreement with the known change in Ca^{2+} concentration in mammalian cells upon histamine stimulation observed with gold standard Ca^{2+} indicators such as Fura-2 (ref. 19).”

Again, perhaps a membrane-localized interaction such as GPCR-mediated recruitment of beta-arrestin could be added to bolster this aspect?

#10 We fully agree that GPCR-mediated recruitment of beta-arrestin could be an interesting process to monitor with splitFAST, and we thank reviewer 2 for this suggestion. However, considering the number of interactions already studied in the paper, we do not believe that such addition is indispensable and meaningful enough for justifying the resubmission delay that would be caused by performing these new experiments.

3) A general concern of all such tagged proteins is do they retain function when engineered with tags such as splitFAST. Can the authors comment on this?

#11 As true for all tag-based methods, we cannot exclude that fusion to CFAST and NFAST can affect in some case the function of the proteins they are fused to. We thus recommend to systematically and carefully verify how fusions to CFAST and NFAST affect protein function. A sentence has been added in the reshaped conclusion to specify this point.